# Human Exposure to Toxic Elements Through Meat Consumption in Africa: A Comprehensive Review of Scientific Literature

**DOI:** 10.3390/nu17111755

**Published:** 2025-05-22

**Authors:** Jose L. Domingo

**Affiliations:** Universitat Rovira i Virgili, Laboratory of Toxicology and Environmental Health, School of Medicine, 43201 Reus, Spain; joseluis.domingo@urv.cat

**Keywords:** toxic elements, meat, dietary exposure, risk assessment, Africa

## Abstract

While meat consumption trends show decreases in some high-income countries, significant increases are observed elsewhere. Although this includes African nations, the average meat consumption in Africa remains generally lower than in many other continents, though patterns vary regionally. Meat provides essential nutrients, but inadequate consumption can pose health problems, while consumption also carries risks including potential exposure to environmental contaminants. This comprehensive review focuses on the recent scientific literature (published 2000–2024) regarding human exposure to specific toxic trace elements, namely arsenic (As), cadmium (Cd), mercury (Hg), lead (Pb), chromium (Cr, particularly hexavalent chromium, Cr(VI)), and nickel (Ni), through the consumption of meat (muscle tissues, organs, and processed products) in Africa. Limited data exist for many African regions, with most studies from Nigeria. Concentrations of these toxic elements in meat tissues varied significantly, with organs like liver and kidney showing higher levels than muscle tissues. Estimated dietary intakes also varied, with some studies indicating potential health risks from Pb, Cd, and As exceeding safety guidelines in specific contexts. However, meat is generally not the primary dietary source of these elements compared to fish, seafood, or staple crops, though risks are higher in areas near pollution sources like mines or waste sites. This study highlights the need for broader research across Central and North Africa, stricter monitoring of meat from high-risk areas, and standardized methodologies to protect public health.

## 1. Introduction

Meat consumption has increased remarkably since the mid-20th century [1]. However, this global increase varies across continents [2]. In high-income countries like those in the European Union (EU), the United States of America (USA), and Canada, meat consumption, particularly of red meat, has shown signs of decreasing or stabilizing. In contrast, many countries worldwide are experiencing significant increases in meat consumption, often correlated with rising incomes and urbanization [3,4]. China, accounting for approximately 27% of global consumption, is the largest total meat consumer, although its per-capita consumption still generally lags behind that of Western countries [5,6].

This trend may be linked to growing public awareness regarding health implications (e.g., associations between high red and processed meat intake and cardiovascular disease, type 2 diabetes, and certain cancers) [6,7,8,9,10,11,12], environmental concerns related to livestock production [6], and ethical considerations.

Regular meat consumption offers nutritional advantages, as a complete source of high-quality proteins, B vitamins, and essential trace elements (iron (Fe), cobalt (Co), phosphorus (P), selenium (Se), and zinc (Zn)), along with fats [7,8]. However, regular consumption of large amounts of certain meats, particularly red and processed meats, can pose health risks. The high saturated fat content in some meats is linked to increased risks of heart disease, stroke, and diabetes [9,10,11,12]. In 2015, the International Agency for Research on Cancer (IARC) classified processed meat as “carcinogenic to humans” (Group 1) and red meat as “probably carcinogenic to humans” (Group 2A), primarily increasing the risk of colorectal cancer [13,14,15,16,17]. Furthermore, antibiotic resistance stemming from the extensive use of antibiotics in livestock production is a significant public health concern [18,19].

Like other food groups, meat consumption can result in exposure to environmental pollutants. Regarding metals/metalloids, toxic elements such as arsenic (As), cadmium (Cd), lead (Pb), mercury (Hg), chromium (Cr, specifically hexavalent chromium, Cr(VI)), and nickel (Ni) are frequently detected at various levels in food alongside essential trace elements. To align with IUPAC recommendations, the term “toxic trace elements” is used here instead of “heavy metals” to avoid imprecise terminology [20,21]. These toxic elements can enter the animal through various pathways and accumulate in tissues. Sources of contamination for livestock include industrial emissions, mining activities, urban waste, agricultural practices (e.g., use of contaminated fertilizers or pesticides on grazing land), contaminated water sources, and the consumption of contaminated feed (including soil or grass ingestion) [22,23,24,25,26,27,28]. However, based on previous comprehensive reviews, meat (referring to tissues from terrestrial animals in this context, distinct from fish/seafood) is generally not considered the primary source of environmental inorganic and organic contaminants for the general human population globally [8,29,30]. Fish and seafood (including marine species, shellfish, crustaceans) typically contribute the most to human dietary exposure to toxic trace elements like Hg and sometimes As and Cd [31,32,33].

Organizations like the World Health Organization/Food and Agriculture Organization (WHO/FAO), United States Environmental Protection Agency (US EPA), and European Food Safety Authority (EFSA) have established recommended intakes or tolerable exposure limits for trace elements. However, rigorous control and monitoring of toxic element levels in food, including meat, is lacking or inconsistent in many countries, including those where meat consumption is increasing. A recent review examined toxic trace element levels (As, Cd, Hg, Pb, Cr, Ni) in meat across Asian countries, finding significant variation in concentrations and confirming that meat was not the primary contributor to dietary exposure for most elements studied [34].

Meat consumption in Africa varies widely due to cultural, economic, and environmental factors. Overall, average per-capita consumption is lower than in other regions like Europe or the Americas, but demand is growing, driven by population growth, urbanization, and economic development [3,4,35,36,37,38,39]. Access and preferences vary based on local resources (beef, goat, sheep, and poultry are common; pork and bushmeat are significant in certain areas) and economic conditions [37]. Contamination of meat by pollutants, particularly toxic trace elements (As, Cd, Hg, Pb, Cr, Ni), is a growing concern due to the presence of various pollution sources across the continent [22,23,24]. This comprehensive review addresses a critical gap by synthesizing recent (2000–2024) evidence on toxic trace element exposure through meat in Africa, providing novel insights into region-specific risks in a data-scarce continent with unique environmental and socio-economic challenges. By focusing on Africa, where data are scarce and meat consumption is rising, this review provides a basis for future research and informs public health strategies to mitigate risks from dietary exposure to toxic elements.

## 2. Methods

### 2.1. Search Strategy

The PubMed and Scopus databases were used to search for information, chosen for their broad coverage of peer-reviewed international biomedical and life sciences literature. The search, detailed here, considered only articles published in English from 1 January 2000 to 31 December 2024. Search terms included combinations of (‘meat’ OR ‘meat products’) AND (‘dietary exposure’ OR ‘human intake’) AND (‘metals’ OR ‘metalloids’ OR ‘trace elements’ OR ‘arsenic’ OR ‘cadmium’ OR ‘lead’ OR ‘mercury’ OR ‘chromium’ OR ‘nickel’) AND (‘human health risks’ OR ‘risk assessment’) AND (‘Africa’ OR names of individual African countries).

### 2.2. Study Selection and Data Extraction

Following the search, titles and abstracts were screened to identify relevant studies. Duplicates were removed using reference management software (e.g., EndNote). Full texts were retrieved and evaluated against inclusion criteria: studies reporting original quantitative data on the concentration of at least one target toxic element (As, Cd, Hg, Pb, Cr, Ni) in edible meat tissues (muscle, organs) or meat products from African countries and/or estimating associated human dietary exposure or health risks. Exclusion criteria included studies focusing solely on essential elements, lacking quantitative data, or not published in English. The grey literature (e.g., reports, theses) was excluded to ensure scientific rigor. Data extracted included toxic element concentrations, sample sizes, tissue types, analytical methods, and risk assessment outcomes (e.g., Estimated Daily Intake [EDI], Target Hazard Quotient [THQ]). Where reported, quality assurance/quality control (QA/QC) protocols and detection limits were noted, though these were often unspecified. Sample sizes and handling of non-detectable samples were extracted where available, but inconsistencies were common. Intake estimations were recorded as reported, typically based on local consumption surveys or assumed rates (e.g., 100 g/day). Concentrations were assumed to be wet weight (ww) unless specified as dry weight (dw), and no unit conversions were performed due to insufficient moisture content data. These limitations are discussed in Section 4.

### 2.3. Health Risk Benchmarks

To contextualize reported concentrations and intakes, health risk benchmarks for toxic trace elements were compiled from WHO, EFSA, and US EPA guidelines (Table 1). These include Provisional Tolerable Weekly Intake (PTWI), Benchmark Dose Lower Limit (BMDL), and Reference Dose (RfD) values, used to assess the significance of dietary exposures. Intakes reported in studies are meat-specific unless otherwise stated.

## 3. Results

### 3.1. Overview

Across the reviewed studies, toxic trace element concentrations in meat varied widely, with Cd and Pb most frequently reported. Approximate ranges (µg/g ww, where specified) included As (0.055–5.32), Cd (0.02–1.35), Hg (0.007–0.12), Pb (0.040–18.90), Cr (1.032–19.37), and Ni (0.20–14.96). Organs (liver, kidney) consistently showed higher concentrations than muscle, with up to 50–60% of samples exceeding national standards for Cd and Pb in some studies (e.g., Ghana [45], Nigeria [46]). Figure 1 visualizes mean Cd and Pb concentrations by country, highlighting regional variability. Data were predominantly from Nigeria (60% of studies), with limited representation from Central and North Africa.

### 3.2. Concentrations of Toxic Trace Elements in African Countries

#### 3.2.1. Nigeria

Nigeria is the country with most data available regarding toxic metal/metalloid levels in meat. While most studies focused on cadmium (Cd) and lead (Pb), data for other elements have also been reported. Ihedioha and Okoye [47] measured Cd and Pb concentrations in muscle and edible offal (kidney, intestine, and tripe) of cows in Enugu State. Mean Cd levels ranged from 0.24 to 0.44 µg/g, detected in 43% of muscle samples and varying percentages of offal samples (100% of kidney, 95% of liver, 70% of intestine, and 50% of tripe). Mean Pb levels ranged from 0.09 to 0.26 µg/g, detected in 70% of muscle and 100% of offal samples. Cd concentrations were relatively high, while Pb concentrations were moderate. In a subsequent study, Ihedioha and Okoye [48] assessed Cd and Pb exposure through the consumption of cow tissues by the Enugu State population. Mean Cd levels were 0.35 ± 0.27, 0.44 ± 0.27, 0.24 ± 0.26, 0.29 ± 0.33, and 0.41 ± 0.33 µg/g (dry weight, dw) for muscle, kidney, liver, intestine, and tripe, respectively. Cd intakes for adult men were 0.23, 0.45, 0.15, 0.55, and 0.50 µg/kg body weight (bw)/week, respectively. Mean Pb concentrations were 0.09 ± 0.16, 0.13 ± 0.07, 0.26 ± 0.25, 0.17 ± 0.12, and 0.17 ± 0.16 µg/g dw for muscle, kidney, liver, intestine, and tripe, respectively, with intakes of 0.89, 0.42, 0.50, 0.94, and 1.21 µg/kg bw/week for adult men. Target hazard quotients (THQs) ranged from 0.42 to 0.90 for Cd and 0.05 to 0.10 for Pb, all < 1, indicating no significant health risks for consumers in that region (Table 2). Further analysis by Ihedioha et al. [49] on the samples of cow meat assessed zinc (Zn), chromium (Cr), and nickel (Ni), calculating mean Estimated Daily Intakes (EDIs) of 299 (Zn), 88.9 (Cr), and 0.76 (Ni) µg/kg bw/day; only Cr intake exceeded the recommended daily intake (RDI), suggesting a potential concern. Meanwhile, Olusola et al. [50] analyzed frozen chicken (thighs and wings) from Lagos and Ibadan, reporting mean Cd (0.0065–0.0078 µg/dL) and Pb (0.0207–0.0227 µg/dL) levels, stating they were within the Nigerian limits. In another investigation, Adetunji et al. [46] measured Cd and Pb concentrations in cattle samples (muscles, liver, and kidney) from Ogun State. Mean Cd concentrations in muscle, liver, and kidney were 0.156, 0.172, and 0.197 µg/g, respectively. Mean Pb levels were 0.721, 0.809, and 0.908 µg/g, respectively. Cd concentrations were within Nigerian standards, while Pb levels exceeded standards in all tissues. Furthermore, Adejumo et al. [51] assessed metals in cured meat products from Southwest Nigeria; they did not detect Pb, Cr, or Ni, but found Cd levels in corned beef and meaty sausage ranging from 0.35 to 1.20 µg/g (mean 0.76 µg/g), exceeding the Nigerian limit. Related to specific pollution sources, Orisakwe et al. [52] studied livestock near a Pb-polluted goldmine in Zamfara State, noting concern about potential Cd and Pb levels in meats based on high levels found in animal blood (Pb up to 7.75 µg/g, Cd up to 0.32 µg/g). Similarly focusing on the environmental context, Ogbomida et al. [53] assessed risks from consuming free-range animals near municipal solid waste sites in Benin City, finding arsenic (As) concentrations up to 0.081 µg/g wet weight (ww) (chicken), Cd up to 0.890 µg/g ww (chicken kidney), Pb up to 0.588 µg/g ww (chicken kidney), and mercury (Hg) up to 0.034 µg/g ww (chicken liver); they highlighted higher concentrations in organs versus muscle and suggested potential risks from Cd and As with prolonged consumption of contaminated chicken liver. On the other hand, Njoga et al. [54] evaluated As, Cd, and Pb in goat carcasses in Enugu State, reporting mean ranges (µg/g) of 0.45–0.57 (As), 0.02–0.06 (Cd), and 0.45–0.82 (Pb) across muscle, liver, and kidney; although EDIs exceeded recommendations, the combined Hazard Index (HI) was <1. Finally, extensive risk assessments by Okoye et al. [55,56] in the Niger Delta found potential risks of Pb exposure (vs. BMDL_0.1_) in children/seniors from goat/cow meat, and significant contributions of meat consumption to dietary As intake (potentially exceeding BMDL_0.1_ for children) and Cd intake (especially in adolescents).

#### 3.2.2. Egypt

Kamaly and Sharkawy [57] measured multiple elements in chicken from Assiut city markets, finding Cd concentrations up to 0.104 µg/g (liver) and Pb levels ranging from 0.146 µg/g (liver) to a high of 5.552 µg/g (chest muscle); calculated EDIs suggested a low risk of Cd but potential concern about Pb, especially from chest meat. Additionally, Mohamed et al. [58] determined As, Cd, Hg, and Pb levels in chilled and frozen beef from the Sharkia Governorate, reporting average As concentrations of 4.66 µg/g (chilled) and 5.32 µg/g (frozen); while EDIs for Cd, Hg, and Pb were below Reference Doses (RfDs), the estimated As intake exceeded the As RfD by 46.7–60%, indicating a potential health concern from As exposure via this beef.

#### 3.2.3. Ghana

Adei and Forson-Adaboh [45], analyzing liver tissues from various animals in Accra and Kumasi, found over 50% of samples exceeded Ghanaian limits for Cd (0.5 µg/g) and Pb (0.5 µg/g). In turn, Bortey-Sam et al. [59] assessed metals in free-range animals near gold mines in Tarkwa, detecting As up to 0.14 µg/g ww (chicken kidney), Cd at its highest in chicken kidney (mean 0.73 µg/g ww), and Hg at its highest in chicken kidney (0.12 µg/g ww) and liver (0.11 µg/g ww); they noted organ accumulation and expressed concern over high Hg levels in free-range chicken exceeding Ghanaian limits.

#### 3.2.4. Uganda

Kasozi et al. [60] measured Cd and Pb in beef, finding no detectable Cd but a high mean Pb concentration (18.90 µg/g); the estimated Pb intake was deemed unsafe for children (high THQ/HI). More recently, Kasozi et al. [61] carried out another beef study, reporting mean levels (µg/g) of 0.41 (Cd), 19.37 (Cr), 14.96 (Ni), and 5.42 (Pb); the calculated EDIs for Cr, Ni, and Pb were higher than WHO Tolerable Daily Intakes (TDIs), suggesting potential risks. A subsequent survey by Kasozi et al. [62] found similar mean levels in beef: 0.4 (Cd), 19 (Cr), 15 (Ni), and 5.5 (Pb) µg/g, with the authors suggesting that high Cr and Ni might be ubiquitous in the sampled beef, possibly from environmental sources.

#### 3.2.5. Other African Countries

*Ethiopia:* A systematic review by Mengistu [63] covering studies from 2016 to 2020 reported mean concentrations in Ethiopian meat/milk (µg/g or similar units) ranging at 0.79–2.96 (As), 1.032–2.72 (Cr), 0.233–0.72 (Cd), and 1.32–3.15 (Pb), noting that levels often exceeded limits, posing potential health risks.*Algeria:* Benamirouche et al. [64] measured Hg and Pb in broiler parts, finding the highest Pb in liver (0.480 µg/g) and highest Hg in breast (0.007 µg/g); calculated EDIs for Hg and Pb exceeded tolerable intakes, and risk assessment suggested potential carcinogenic risks from Pb.*Senegal:* Missohou et al. [65] linked Hg contamination in poultry meat (>0.011 µg/g in 20% of samples) to a nearby landfill impacting well water used for poultry.*South Africa*: Ambushe et al. [66] investigated bovine tissues from mining areas, confirming differential tissue accumulation, with the highest Cd (1.35 µg/g) and Pb (0.62 µg/g) found in bone, but also high levels in kidney and liver compared to muscle.*Mauritania:* Ahmed et al. [67] analyzed dromedary meat, reporting mean toxic element levels (µg/g) of 0.055 (As), 0.064 (Cd), 0.027 (Hg), and 0.040 (Pb), recommending further monitoring, especially of edible organs.

Table 3 summarizes the levels of toxic trace elements (As, Cd, Hg, Pb, Cr, Ni) in meat and meat products from African countries (excluding Nigeria) based on studies published between 2000 and 2024, with added details on sample sizes, QA/QC, detection limits, and intake estimation methods.

## 4. Discussion

Meat consumption varies considerably between countries depending on various factors. Africa is no exception, as meat consumption varies significantly based on income levels, cultural practices, urbanization, and the availability of meat. Africa generally has a lower average per-capita meat consumption compared to other regions like Europe or North America, but demand has increased in recent decades due to population growth, economic development, and urbanization [38,39]. The animal species most consumed in African countries include beef, goat, sheep, poultry, pork, and sometimes bushmeat.

Various pollution sources can contaminate meat with environmental pollutants, including toxic trace elements [29,68,69]. Sources relevant to Africa include mining operations, industrial discharges, agricultural runoff (pesticides, fertilizers), improper waste disposal (municipal and e-waste dumpsites), contaminated irrigation water, and the use of contaminated animal feed or grazing on contaminated land [25,26,27,28]. Toxic trace elements such as As, Cd, Hg, Pb, Cr, and Ni can be taken up by livestock from these sources and subsequently accumulate in their tissues, particularly in organs like the liver and kidneys, potentially affecting the safety of the meat consumed by humans [20,23,52,70]. This review’s findings align with global studies, such as those in Asia [34], showing that meat is generally not the primary dietary source of toxic elements compared to fish or crops, but our data highlight specific African contexts where meat contributes significantly to exposure, particularly for Cd and Pb in Nigeria [47,48,54], and As in Egypt [58]. For instance, studies in Nigeria reported Cd levels in cow offal (up to 0.44 µg/g) and Pb in goat carcasses (up to 0.82 µg/g), with some intakes exceeding safety thresholds (Table 1), similar to concerns raised in Ghana [45,59] for Cd and Hg near mining areas. In contrast, our results suggest lower risks in areas like Mauritania [67], where mean levels were below 0.1 µg/g for all elements.

The higher accumulation of toxic elements in organs versus muscle, a consistent finding in the current review [45,47,52,58,66], closely reflects global patterns of tissue-specific metal bioaccumulation reported in the literature [8,29,34]. For example, Cd levels in chicken kidney (0.89 µg/g in Nigeria [53]) and Pb in beef liver (0.62 µg/g in South Africa [66]) were notably higher than in muscle tissues, aligning with studies in Asia [34] reporting elevated Cd in offal. This suggests that consumption of organ meats in Africa, particularly in high-risk areas like mining regions, may pose greater risks than muscle consumption. Unlike fish-heavy diets in coastal regions globally [31,32], where Hg and As dominate exposure, our review indicates that in African inland areas, Pb and Cd from meat are more pressing concerns due to local pollution sources like mines and waste sites [52,59].

The significant number of studies from Nigeria highlights a potential publication bias or greater research activity in that country, limiting continent-wide generalizations. Data from Central and North Africa are particularly scarce, underscoring the need for broader geographical coverage. Variations in sampling strategies, analytical methods, and reporting units (e.g., ww vs. dw) further complicate comparisons, as unit conversions were not feasible due to missing moisture content data. Additionally, inconsistent reporting of QA/QC protocols, detection limits, and sample sizes reduces data reliability, though excluding such studies would eliminate most African data [45,47,53]. These limitations highlight the need for standardized methodologies in future research.

Differences in animal sources (e.g., industrial farming vs. free-range grazing) likely influence contaminant levels, as free-range animals near pollution sources (e.g., mines, waste sites) may accumulate higher levels of toxic elements [52,59]. However, many studies do not specify the animal origin, limiting direct comparisons. Future research should clarify whether meat is from industrial, wild, or imported sources to better assess exposure risks.

High-risk groups, such as children and women of childbearing age, may face greater health impacts from toxic element exposure. Studies in Nigeria [55,56] and Uganda [60] reported elevated Pb and As intakes in children, exceeding BMDL_0.1_ values, posing risks for neurotoxicity and cancer. Women of childbearing age may be vulnerable to Cd and Hg accumulation, which can affect fetal development, though data are limited. Targeted risk assessments for these groups are needed to inform public health strategies.

Emerging research suggests that nutraceuticals (e.g., flavonoids) and functional foods may mitigate PTE-induced organ damage, potentially via microbiome interactions [71,72]. While promising, these interventions are beyond the scope of this review but warrant further study in African contexts with high meat consumption.

Strategies to reduce potential exposure through meat could include stricter control of environmental pollution from industrial, mining, and waste sources; monitoring animal feed and water quality; targeted surveillance of meat (especially organ meats) from high-risk areas; and potentially consumer advice on dietary diversity and limiting frequent consumption of organs known to accumulate contaminants. These strategies are particularly urgent in regions like Nigeria and Uganda, where the present review found elevated Pb and Cd levels [54,60,61], and align with recommendations from global studies [24,34] advocating for integrated pollution control and dietary risk communication.

A general trend observed globally, which should also apply to Africa, is the notable reduction in environmental Pb contamination over recent decades. This is primarily attributed to the successful international phase-out of leaded gasoline and restrictions on lead-based paints, which has lowered background Pb levels in soil, water, and the food chain [20]. However, our findings indicate that localized Pb hotspots, such as mining areas in Nigeria [52] and Uganda [60], remain a concern, suggesting that Africa-specific interventions are needed to address these persistent sources, unlike the broader global decline in Pb exposure.

## 5. Conclusions

Synthesizing evidence from 2000–2024, this review highlights key findings on toxic trace element exposure through meat consumption in Africa:Toxic trace elements (As, Cd, Hg, Pb, Cr, Ni) accumulate more in organs (liver, kidney) than muscle, with Cd and Pb posing significant risks in Nigeria, Egypt, Ghana, and Uganda, where intakes occasionally exceed WHO/EFSA benchmarks (e.g., Pb BMDL_0.1_).Meat is generally a secondary contributor to dietary exposure compared to fish or crops, but risks are elevated near pollution sources like mines or waste sites.Data are heavily skewed toward Nigeria, with sparse coverage in Central and North Africa, limiting continent-wide conclusions and suggesting potential bias.Research gaps include limited data on Hg and Ni, inconsistent reporting of QA/QC and detection limits, and lack of standardized risk assessments. Future studies should prioritize longitudinal exposure analyses, multi-element studies, and vulnerable populations (e.g., children, women of childbearing age).Continued surveillance of meat, especially organs, from high-risk areas is essential, alongside pollution control and consumer education to mitigate risks as meat consumption rises.

These findings underscore the urgent need for broader, standardized research to protect public health in Africa’s evolving dietary landscape.

## Figures and Tables

**Figure 1 nutrients-17-01755-f001:**
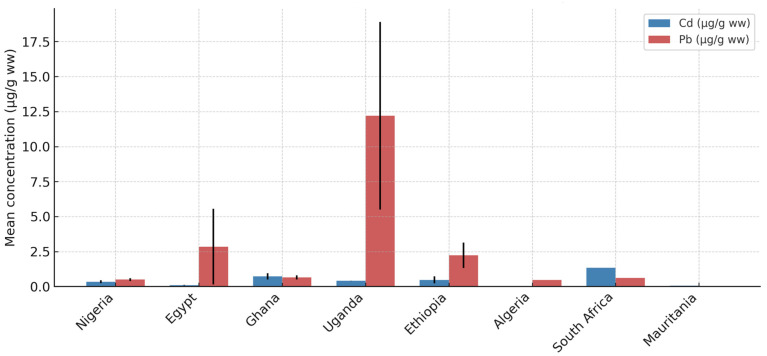
Mean concentrations of Cd and Pb in meat across African countries. Data from studies published in 2000–2024, converted to wet weight where specified. Variability reflects analytical and environmental factors.

**Table 1 nutrients-17-01755-t001:** Health risk benchmarks for toxic trace elements.

Element	Benchmark/Parameter	Value	Source
As	BMDL_0.1_ (cancer)	0.3–8 µg/kg bw/day	EFSA [40]
Cd	PTWI	25 µg/kg bw/week	WHO [41]
Hg	PTWI	4 µg/kg bw/week	WHO [41]
Pb	BMDL_0.1_ (neurotoxicity)	0.5 µg/kg bw/day	EFSA [42]
Cr (VI)	RfD	3 µg/kg bw/day	US EPA [43]
Ni	TDI	2.8 µg/kg bw/day	EFSA [44]

BMDL_0.1_: Benchmark Dose Lower Confidence Limit for a 0.1% response; PTWI: Provisional Tolerable Weekly Intake; RfD: Reference Dose; TDI: Tolerable Daily Intake; bw: body weight.

**Table 2 nutrients-17-01755-t002:** A summary of studies conducted in the current century in Nigeria on the levels of toxic trace elements in edible meat and meat products.

Region/City	Analyzed Meat/Meat Products	Results for the Analyzed Toxic Elements	Remarks	References
Enugu State	Muscle and edible offal (kidney, intestine, and tripe) of cows	Mean Cd levels (range): 0.24–0.44 μg/g. Mean Pb levels (range): 0.09–0.26 μg/g.	The highest intakes of Cd and Pb were 0.55 µg/kg bw/week (intestine) and 1.21 µg/kg bw/week (tripe), respectively.	Ihedioha and Okoye [47,48]
Lagos and Ibadan	Frozen chicken (thighs and wings)	Mean concentrations: 0.0065 and 0.0078 µg/dL for Cd, and 0.0207 and 0.0227 µg/dL for Pb.	The Cd and Pb concentrations were within the maximum residue levels allowed in the country.	Olusola et al. [50]
Ogun State	Cattle (muscles, liver, and kidney)	The mean levels of Cd in muscle, liver, and kidney were 0.156, 0.172, and 0.197 µg/g, respectively. For Pb, the mean levels were 0.721, 0.809, and 0.908 µg/g for muscle, liver, and kidney, respectively.	The levels of Pb were higher than the standards in all bovine tissues.	Adetunji et al. [46]
Nigeria (area/region not specified)	Muscle and edible offal of cows	The minimum and maximum mean levels (µg/g ww) were 1.24 (muscle) and 4.28 (liver) for Cr, and 0.20 (liver) and 0.36 (kidney) for Ni.	The mean EDIs were 88.9 and 0.76 µg/kg bw/day for Cr and Ni, respectively. Only the intake of Cr was higher than the recommended daily intake.	Ihedioha et al. [49]
Southwest of the country	Cured meat products (ham, bacon, sausages, corned beef, and luncheon)	Pb, Cr, and Ni were not detected in any sample, while Cd levels ranged between 0.35 and 1.20 μg/g in samples of corned beef and meaty sausage, respectively, with 0.76 μg/g its mean level.	Cd concentrations exceeded the maximum allowable limit.	Adejumo et al. [51]
Dareta and Abare, Zamfara State (around a lead-polluted goldmine)	Goat, sheep, cattle, and chicken	The highest mean levels of Pb and Cd were found in samples of goat and chicken (blood), with values of 7.75 and 0.32 μg/g, respectively.	The levels of Cd and Pb in meats of the examined area would be of concern.	Orisakwe et al. [52]
Benin City (near municipal solid waste sites)	Muscle, liver, kidney, and gizzard of free-range animals (chicken, cattle, and goats)	The highest concentrations were 0.081 μg/g for As in chicken muscle, and 0.890 and 0.588 μg/g for Cd and Pb, respectively. The highest Hg levels corresponded to chicken liver (0.034 μg/g) and kidney (0.030 μg/g).	Potential health risks were suggested for those individuals with continuous exposure to As and Cd by consumption of contaminated meats.	Ogbomida et al. [53]
Enugu State	Muscle, liver, and kidney from goat carcasses	The ranges of the mean concentrations (μg/g) were the following: As, 0.45 (muscle)–0.57 (liver); Cd, 0.02 (muscle and liver)–0.06 (kidney), and Pb, 0.45 (liver)–0.82 (muscle).	The hazard index (HI) was <1 for As, Cd, and Pb.	Njoga et al. [54]
Six areas of the Niger Delta	Goat, chicken, and cow	As and various metals (Cd, Cu, Hg, Pb, V, and Zn) were measured in meat samples of the indicated animals’ species. Numerous data were obtained and classified by age groups of the population.	No potential health hazards were found for Hg and V. The risk for Pb exposure was above (or close to) BMDL_0.1_ for developmental neurotoxicity and nephrotoxicity (intake of meats of goat and cow, respectively) in children and seniors, in the six areas included in the survey. Meat consumption also contributed to the dietary intake of As (especially for children, exceeding the BMDL_0.1_) and Cd (especially in the group of adolescents).	Okoye et al. [55,56]

**Table 3 nutrients-17-01755-t003:** A summary of studies conducted in the current century in various African countries (excepting Nigeria) on the levels of toxic trace elements in edible meat and meat products.

Country/Region	Analyzed Meat/Meat Products	Results for the Analyzed Toxic Elements	Remarks	References
Egypt (Assiut city)	Chicken (muscle, liver)	Mean Cd: up to 0.104 µg/g (liver). Mean Pb: 0.146 µg/g (liver) to 5.552 µg/g (chest muscle).	EDIs suggested low risk for Cd but potential concern for Pb, especially from consumption of chest meat.	Kamaly and Sharkawy [57]
Egypt (Sharkia Governorate)	Chilled and frozen beef	Mean As: 4.66 µg/g (chilled), 5.32 µg/g (frozen). Cd, Hg, Pb below RfDs.	As intake exceeded RfD by 46.7–60%, indicating potential health concern.	Mohamed et al. [58]
Ghana (Accra, Kumasi)	Liver tissues (various animals)	Cd: >0.5 µg/g in over 50% of samples. Pb: >0.5 µg/g in over 50% of samples.	Exceeded Ghanaian limits for Cd and Pb.	Adei and Forson-Adaboh [45]
Ghana (Tarkwa, near gold mines)	Free-range animals (chicken kidney, liver)	Mean As: up to 0.14 µg/g (kidney). Mean Cd: 0.73 µg/g (kidney). Mean Hg: 0.12 µg/g (kidney), 0.11 µg/g (liver).	High Hg levels in chicken exceeded Ghanaian limits, with organ accumulation noted.	Bortey-Sam et al. [59]
Uganda (Southwestern)	Beef	Cd: not detected. Mean Pb: 18.90 µg/g.	Pb intake unsafe for children (high THQ/HI).	Kasozi et al. [60]
Uganda (Eastern)	Beef	Mean Cd: 0.41 µg/g. Mean Cr: 19.37 µg/g. Mean Ni: 14.96 µg/g. Mean Pb: 5.42 µg/g.	EDIs for Cr, Ni, Pb exceeded WHO TDIs, suggesting potential risks.	Kasozi et al. [61,62]
Ethiopia	Meat	Mean As: 0.79–2.96 µg/g. Mean Cr: 1.032–2.72 µg/g. Mean Cd: 0.233–0.72 µg/g. Mean Pb: 1.32–3.15 µg/g.	Levels often exceeded limits, posing potential health risks.	Mengistu [63]
Algeria	Broiler parts (liver, breast)	Mean Pb: 0.480 µg/g (liver). Mean Hg: 0.007 µg/g (breast).	EDIs for Hg and Pb exceeded tolerable intakes.	Benamirouche et al. [64]
Senegal (near Mbeubeuss landfill)	Poultry meat	Hg: >0.011 µg/g in 20% of samples.	Linked to landfill-contaminated well water used for poultry.	Missohou et al. [65]
South Africa (mining areas)	Bovine tissues (bone, kidney, liver, muscle)	Mean Cd: 1.35 µg/g (bone). Mean Pb: 0.62 µg/g (bone). High levels in kidney, liver vs. muscle.	Confirmed differential tissue accumulation.	Ambushe et al. [66]
Mauritania	Dromedary meat	Mean As: 0.055 µg/g. Mean Cd: 0.064 µg/g. Mean Hg: 0.027 µg/g. Mean Pb: 0.040 µg/g.	Low levels; further monitoring recommended, especially for edible organs.	Ahmed et al. [67]

## Data Availability

Data are contained within the article (as it is a review summarizing published data cited in the references).

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
