# Peer review of "Human Exposure to Toxic Elements Through Meat Consumption in Africa: A Comprehensive Review of Scientific Literature"

_nutrients, 2025, doi:10.3390/nu17111755_

Round 1

Reviewer 1 Report

Comments and Suggestions for Authors

My review comments are as follows:

In Methods:

Only the use of PubMed and Scopus, keywords, inclusion and exclusion criteria were mentioned. It does not explain the details of the screening process, whether duplicates were excluded, or whether quality assessment was performed.

It is recommended to supplement the basic process or literature screening instructions.

Some research reports have proposed THQ or EDI, but this article does not compile reference health risk benchmark values ​​for each metal, such as WHO's PTWI and EFSA's BMDL. Recommendations should be summarized and presented in the Methods or Discussion to help readers understand the relative significance of each metal concentration and risk.

In Results:

Although data are collated in Table 1 (Nigeria) and Table 2 (other African countries), they are mostly in list format and lack statistical integration (such as mean, median, and percentage exceeding the standard). It is recommended to present it in a visual design (such as bar chart, heat map).

A few references clearly marked the units (such as Ihedioha et al. using dw and Ogbomida et al. using ww), but most did not fully indicate them and were not marked in the tables.

The methods section also does not specify whether unit conversion or unification was performed. It is recommended to treat it uniformly or note this limitation.

The author also mentioned in lines 249–251 that the units used in different documents are not consistent.

In Discussion:

The animal sources in various studies include cattle, chickens, pigs, etc., but some literature does not clearly distinguish the source type, such as whether it comes from industrial areas, wild areas, or imported. This article also does not discuss the differences in pollution levels that may be caused by differences in sources. It is recommended to add relevant information.

The health risks for the general population have been mentioned, it is recommended that risk estimation be conducted for high-risk groups such as children and women of childbearing age. It also explains that if these groups consume the same meat, the impact on health risks may be more significant.

In Conclusions:

The conclusions focus on increased monitoring and risks in highly polluted areas, but do not specifically identify remaining research gaps in the current data. For example, there is little data on Central Africa and a lack of quantitative risk assessment for certain metals. It is recommended that authors add brief suggestions to enhance the academic extension and practicality of the article.

Others:

The author also acknowledges that there is a disproportionate amount of data from Nigeria, which may lead to bias. However, there is no extended discussion of its impact on the overall conclusions in the conclusion or recommendation paragraphs.

Author Response

Attached please find a file containg the detailed responses to the comments and suggestions of this Reviewer (1).

Reviewer 2 Report

Comments and Suggestions for Authors

In the present review article, Jose L. Domingo focused on recent scientific literature (published 2000-2024) regarding human exposure to specific toxic trace elements, namely arsenic (As), cadmium (Cd), mercury (Hg), lead (Pb), chromium (Cr, particularly hexavalent chromium, Cr(VI)), and nickel (Ni), through the consumption of meat (muscle tissues, organs, and processed products) in Africa.

So far, this study highlights the need for broader research across Africa and stricter monitoring of meat from high-risk areas to protect public health. Overall, I think that the paper could be of interest to the readers of "Nutrients" and researchers, in general.

However, I raise a series of crucial points to address carefully for improve, in my humble opinion, the overall quality of manuscript.

1) Please clarify the type of review (i.e. scoping, systematic, narrative) in the title and methods of paper. Of course, this aspect is crucial in order to reliably determine health and medical practice standards and public policy in Africa on the basis of data here analyzed.

2) Indeed, if this paper is a “scoping review”, it may be appropriate to register the present review in a public register (for example, Research registry, Open Science, JBI etc.) where the author further certify the compliance of this review with PRISMA guidelines.

3) Although the term "heavy metals" is still widely used, its use was in fact disapproved by IUPAC (Duffus J.H., 2002) and later recommended to eventually be replaced by "potentially toxic elements" (PTEs) if not with "elements" only (Pourret O et al., 2019). To be consistent, we should only use well-accepted definitions.

Duffus, J.H. Heavy Metal - A Meaningless Term? (IUPAC Technical Report). Pure Appl. Chem. 2002, 74, 793-807.

Pourret, O.; Hursthouse, A. It's Time to Replace the Term "Heavy Metals" with "Potentially Toxic Elements" When Reporting Environmental Research. Int. J. Environ. Res. Public Health. 2019, 16, 4446.

4) There is recent evidence that nutraceuticals and/or antioxidants/antinflammatory compounds such as Inositols or Flavonoids showed a benefit against organ alterations caused by exposure to PTEs. So, considering the current scientific knowledge, synergistic effects of nutraceuticals with different medical therapy properly combined with good agricultural practice and healthy eating habits could represent a definite strategy to counteract PTESs toxicity (as Cd, for example). Please discuss this intriguing topic (also from a translational perspective) and for your convenience you could consider the following references (Metabolites 2023, 13(6):722; Int J Environ Res Public Health. 2022, 19(19):12380; Bull Environ Contam Toxicol. 2021, 106(1):65-74; Curr Med Chem. 2017, 24(35):3879-3893) in your paper.

5) Functional foods, nutraceuticals, or dietary supplements in the context of PTEs-induced organ damage, should address the impact on the microbioma and all potential interactions with a preventive and/or therapeutic intervention. Please discuss this intriguing topic in the revised version of paper (see for your convenience: Br. J. Pharmacol. 2020, 177, 1351-1362).

Author Response

Attached please find a file containing the responses to the comments and suggestions of this Reviewer (2).

Reviewer 3 Report

Comments and Suggestions for Authors

This manuscript is a review article on toxic metal/metalloid levels in, and intakes from, a variety of meat in some African countries. The review was quite descriptive and the author has simply listed the literature data with no attempt to synthesize them to draw some relevant new insight. The “findings” of this review, e.g., internal organ contains more metal than muscle tissue, meat is not the predominant food of dietary metal intake, or meat from pollution source can contain high levels of metal, are not novel, unfortunately. Some of the local readers might be interested in this review article but international readers might not.

As a descriptive review article, the following information must be provided.

  1. Clearly specify the reported concentrations were dry weight basis or wet basis for each of the literature.
  2. Clearly indicate number of samples for each type of meat analyzed in each of the literatures.
  3. Clearly indicate whether or not extensive internal and external QA/QC of element analysis was reported in each of the literatures. Ideally, only the literature clearly declares QA/QC should be included in the review. No declaration of QA/QC indicates non-reliable data: high metal concentration in meat in such literature may be due to analytical contamination or some other inadequacies. Only the reliable data should be included in the review.
  4. Detection limit of the metal must be presented for each of the literature.
  5. Clearly specify how the mean value has been calculated when non-detectable sample was present in the data for each of the literatures. It must be noted that the non-detectable samples MUST BE included for the calculation of the mean: the literature in which non-detectable sample(s) has not been included should be removed from the review.
  6. Specify the method of estimation of daily metal intake from meat for each of the literature: how the consumption rate of the meat was estimated.
  7. Clearly specify if the estimated intake for the comparison with RfD or TDI was an intake from meat only or intake from all of the food including meat.

If even one of information is missing in the literature, then the literature should not be included in the review because the literature does not provide reliable data.

In addition, CONCLUSION section of this manuscript is too lengthy. Conclusion must be concise.

Author Response

Attached please find a file containing the responses to the comments and suggestions of this Reviewer (3).

Round 2

Reviewer 1 Report

Comments and Suggestions for Authors

The author has corrected or supplemented the information in the manuscript and responded to my comments and suggestions in detail.

Reviewer 2 Report

Comments and Suggestions for Authors

Thank you for addressing my comments well. I have no further remarks.